# The Influence of the Seasonal Variability of *Candida* spp. Bloodstream Infections and Antifungal Treatment: A Mediterranean Pilot Study

**DOI:** 10.3390/antibiotics14050452

**Published:** 2025-04-29

**Authors:** Paola Di Carlo, Nicola Serra, Ornella Collotta, Claudia Colomba, Alberto Firenze, Luigi Aprea, Salvatore Antonino Distefano, Andrea Cortegiani, Giovanni Giammanco, Teresa Maria Assunta Fasciana, Roberta Virruso, Angela Capuano, Consolato M. Sergi, Antonio Cascio

**Affiliations:** 1Department of Health Promotion, Mother and Child Care, Internal Medicine and Medical Specialties, University of Palermo, 90127 Palermo, Italy; paola.dicarlo@unipa.it (P.D.C.); claudia.colomba@unipa.it (C.C.); alberto.firenze@unipa.it (A.F.); giovanni.giammanco@unipa.it (G.G.); teresa.fasciana@unipa.it (T.M.A.F.); antonio.cascio03@unipa.it (A.C.); 2Department of Neuroscience, Reproductive Sciences and Dentistry Department, Audiology Section “Federico II” University of Naples Federico II, 80131 Naples, Italy; 3Degree Course in Medicine and Surgery, Medical School of Hypatia, University of Palermo, 93100 Caltanissetta, Italy; ornellacollotta96@gmail.com; 4Azienda Ospedaliera Universitaria Policlinico “Paolo Giaccone”, 90127 Palermo, Italy; luigi.aprea@policlinico.pa.it (L.A.); salvatoreantonino.distefano@policlinico.pa.it (S.A.D.); roberta.virruso@policlinico.pa.it (R.V.); 5Department of Precision Medicine in Medical, Surgical and Critical Care Area (Me.Pre.C.C.), University of Palermo, 90127 Palermo, Italy; andrea.cortegiani@unipa.it; 6Department of Emergency, AORN Santobono-Pausilipon, 80122 Naples, Italy; a.capuano@santobonopausilipon.it; 7Anatomic Pathology Division, Children’s Hospital of Eastern Ontario, University of Ottawa, Ottawa, ON K1H 8M5, Canada; csergi@cheo.on.ca; 8Department of Laboratory Medicine and Pathology, University of Alberta, Edmonton, AB T6G 2R3, Canada

**Keywords:** seasonality, candidemia, survey, antifungal, *Candida parapsilosis*, *Candida glabrata*, *Candida albicans*, fluconazole

## Abstract

**Background/Objectives**: Various factors associated with seasonality, including temperature, humidity, geographical composition, and seasonal fluctuations, can influence the trends of microbes responsible for hospital infections, such as *Candida* spp. This study evaluates the seasonal variability of *Candida* spp. bloodstream infections and antifungal resistance in hospitalized patients in Sicily. **Methods**: We retrospectively analyzed the demographic and epidemiological characteristics of 175 patients with blood cultures positive for *Candida* spp. Who were hospitalized at University Hospital Paolo Giaccone (A.U.O.P.), University of Palermo, Italy, from 1 January 2022 to 31 December 2024. Data on *Candida* species and antifungal resistance were also collected from the hospital’s database system to prevent and control hospital infections in A.U.O.P. **Results**: A total of 175 patients, 57.7% males, with a mean age of 68.3 years, were included in this study. *Candida parapsilosis*, *Candida albicans*, and *Candida glabrata* were more frequent in ICU (54.5%, *p* = 0.0001), medical (72.5%, *p* = 0.0003), and surgical settings (24%, *p* = 0.0161), respectively. *C. parapsilosis* was more frequent in dead patients (53.2%, *p* = 0.005). Among the seasons, we observed a significantly higher presence of *C. glabrata* in Autumn (20%, *p* = 0.0436). From the analysis of the seasons, *C*. *parapsilosis* and *C*. albicans were more frequent for each season, except in Spring, where the most frequent isolates were *C*. *glabrata* (5.1%, *p* = 0.0237) and C. *parapsilosis* (9.7%, *p* < 0.0001). The antifungal with the most resistance to *Candida* spp. was fluconazole in all seasons. **Conclusions**: Our study highlights the seasonal trends in *Candida* spp. and antifungal resistance, emphasizing climate change’s challenges on fungal diseases. These findings may contribute to improving prevention and treatment strategies for candidemia.

## 1. Introduction

In 2022, the World Health Organization (WHO) identified *Candida* species such as *Candida auris* and *C. albicans* as critical-priority pathogens and *C. tropicalis* and *C. parapsilosis* as high-priority pathogens. Despite better diagnostics and available antifungal treatments, hospitalized patients with invasive candidiasis (IC) infections, including candidemia, still face high morbidity and mortality rates. IC is also a common bloodstream infection in Europe, with increased incidence in Italy over the past two decades [1,2,3,4].

Trends in candidemia have changed significantly in recent decades. The species distribution varies among countries. Currently, *Candida albicans* represents the most common *Candida* species in Europe (40% to 60% of cases). *Candida glabrata* is more prevalent in Northern Europe, the most common species in four geographical areas, such as the United Kingdom, France, Slovenia, and Belgium, ranging from 25% to 33%. In contrast, Candida parapsilosis has become increasingly common in Southern Europe, such as in Italy and Turkey, with rates between 24% and 26% [5].

This infection occurs when the yeast enters the bloodstream and has the potential to spread to various organs, including the eyes, heart, brain, and kidneys. Hospitalized patients, particularly those who are critically ill or frail, are at risk for invasive candidiasis. As a common healthcare-associated infection, it is essential to recognize the potential for serious complications and to take preventive measures to protect at-risk patients [3,6,7]. Surgical wounds and invasive medical devices, such as catheters, central venous lines, and drainages, allow *Candida* to enter the body [8,9]. *C. albicans* is recognized as the most prevalent species associated with candidemia, yet the growing list of *Candida* species implicated in this condition reflects the advancements in identification techniques. Moreover, their distribution varies in population-based studies conducted in different geographical areas. The ARTEMIS DISK Global Antifungal Surveillance Study has made significant strides by establishing an extensive registry of invasive *Candida* isolates gathered from 127 medical centers across 39 countries [9]. The trend is variable; traditional key species like *C. albicans*, *C. glabrata*, *C. tropicalis*, *C. parapsilosis*, and *C. krusei* are now accompanied by emerging species, including *C. auris* [10,11].

Seasonal variations are tied to climate factors such as environmental humidity, temperature, atmospheric pressure, and wind, which can impact fungal community composition. As global temperatures rise, fungi adapt, affecting their interactions with other organisms and often leading to diseases. Warmer, wetter conditions allow fungal pathogens to thrive, causing crop failures and increasing human infections, especially in regions with health disparities. These changes demand immediate attention [12,13].

Recent studies such as that of Arendrup, M. C. [5] show the impact of regional climate, reporting different *Candida* species in various European and non-European countries characterized by different geographical climes.

Research indicates that climate change is associated with new fungal pathogens and existing ones appearing on different hosts, some of which exhibit high drug resistance. Studies suggest these pathogens may have recently adapted to endure higher environmental temperatures [14,15,16].

Traditionally, *Candida* spp. sessile cells exhibit a variable response to temperature, with significant differences between strains [17,18]. Its yeast-to-hypha shape conversion also promotes the formation of a biofilm on inert (medical devices like catheters, shunts, and stents) or biological (skin or mucosa) surfaces, making it more harmful [19]. The surrounding environment influences the growth of the biofilms, and although temperature is a known factor that alters the yeast-to-hypha transition in *C. albicans*, its impact on biofilm formation is not yet fully understood [19,20,21]. All persistent *Candida* spp. infections can be successfully treated with standard fungicides like fluconazole, Amphotericin-B, and echinocandins; however, biofilm-related infections are hard to control by standard treatment.

Like many other European countries, Italy has been affected by climate change. Various factors associated with climate change, including temperature, humidity, geographical composition, and seasonal fluctuations, can influence the trends of microbes responsible for hospital infections, such as *Candida* spp. [22,23].

This study evaluates the seasonal variability of *Candida* spp. bloodstream infections and antifungal resistance in hospitalized patients in Sicily.

## 2. Results

Positive blood samples of *Candida* spp. were obtained from 175 consecutive patients with a male/female ratio of 1.33:1 (100 M and 75 F), ages 18–92 years old, a mean age of 68.3 years, and a standard deviation (SD) of 14.2. In Table 1, we report the general characteristics of the 175 study participants.

In Table 1, we also report the comorbidities related to candidemia. Notably, the more frequent comorbidities were chronic heart failure (34.3%, *p* < 0.0001), chronic obstructive pulmonary disease (22.3%, *p* = 0.0018), chronic renal failure (20.0%, *p* = 0.025), diabetes (20.6%, *p* = 0.0138), and organ solid tumors (20.6%, *p* = 0.0138).

Table 2 shows the percentages of *Candida* spp. considering both setting and season.

From Table 2, about *Candida* distribution, no significant relationship between season and hospital setting was observed (*p* = 0.19, by Fisher’s exact test).

Table 3 shows the percentages of *Candida* spp. resistance, considering antifungal and season. In the table, all percentages refer to the 175 patients. Additionally, in the table, we report two statistical analyses. The first analysis is reported in the last column, where we show the analysis among seasons for each antifungal. In contrast, the second analysis is reported in the last row, where we tested among the antifungals used the one with the highest resistance to *Candida* spp. in each season. Particularly, the effect size was calculated for each significant analysis.

From Table 3, we found significantly more resistance to *Candida* spp. in Summer (32%, *p* = 0.0243, post hoc Z-test) than in Autumn (28.6%), Spring (21.1%), or Winter (18.3%). Additionally, for each antifungal, no significant differences in resistance to *Candida* spp. were observed; while considering the antifungals with more resistance in each season, we found that fluconazole was the antifungal with the highest percentage of resistance to *Candida* spp.

From our power analysis, we observed a minimum bias in the statistical analysis due to the small sample size, i.e., the significant results had a low probability of bias effect due to the small sample.

Table 4 shows the distribution of *Candida* spp. according to the socio-demographic and setting variables of Table 1, such as age, gender, patients’ status (dead), and hospital setting (ICU, medical, and surgical).

In Table 4, no significant differences about age and gender for patients with different *Candida* spp. can be observed (*p* = 0.71 and *p* = 0.88, respectively). In contrast, for the status variable, among *Candida* spp., we found more patients died with *C. parapsilosis* (53.2%, *p* = 0.005). Furthermore, among *Candida* spp., patients with *C. parapsilosis*, *C. albicans*, and *C. glabrata* were more frequent in the ICU (54.5%, *p* = 0.0001), medical settings (72.5%, *p* = 0.0003), and surgical settings (24%, 0.0161), respectively.

The minimum biases for all the significance tests in Table 4 were observed via power analysis.

Table 5 shows the *Candida* species found in our sample stratified by season. In Table 5, two analyses are described. The first analysis is reported in the last column, where we show the analysis among seasons for each *Candida* spp., while the second analysis is reported in the last row, showing the *Candida* found more frequently among the *Candida* spp. for each season.

Table 5 shows a significantly higher presence of C. glabrata in Autumn (5.7%, *p* = 0.0481) among the seasons. Additionally, from the analysis of every season among *Candida* spp., we found that *C. parapsilosis* and *C. albicans* were the most frequent in each season, apart from Spring, where the most frequent *Candida* spp. were *C. glabrata* (5.1%, *p* = 0.0237) and *C. parapsilosis* (9.7%, *p* < 0.0001). From our power analysis, we observed a minimum bias in the statistical analysis due to the small sample size; i.e., the significant results had a low probability of bias effect due to the small sample.

In Table 6, we report some characteristics of the patients with positive blood samples for *Candida* spp. with central venous catheters.

From Table 6, we observed in patients with CVCs no significant difference among hospital settings (*p* = 0.062) and comorbidities (*p* = 0.52), while among *Candida* spp., *Candida parapsilosis* was the most frequent (50.0%, *p* < 0.0001).

Finally, in Figure 1, we show the resistance to all the antifungal drugs used in this study, considering all the seasons and only the *Candida* spp. more frequent in Table 2, such as *C. albicans*, *C. glabrata*, *C. parapsilosis*, and *C. tropicalis*. Notably, the table associated with the figure shows the resistance to all the specific antifungals used for each season.

## 3. Discussion

This is the first study that has assessed the seasonal distribution of *Candida* species and antifungal susceptibility in the bloodstream in Southern Italy after the COVID-19 pandemic era. We found a significantly higher prevalence of candidemia from *Candida* spp. in Summer (32%, *p* = 0.0243) than in Autumn (28.6%), Spring (21.1%), and Winter (18.3%). This result follows other studies performed on neonates and women, where *Candida* infection was more common in Summer [24,25].

The significant presence of *Candida* spp. in Summer could be due to the higher temperature and humidity, which create the perfect conditions for yeast, especially for Candida. Some species of *Candida* spp. are thermotolerant, meaning they can grow and survive at higher temperatures, such as 37 °C. Environmental factors, including seasonal climate changes and geographical distribution, can influence Candida’s ability to produce biofilms on synthetic materials. This characteristic enhances the organisms’ adhesion to medical devices, making infections more resistant to treatment [26,27]. Additionally, favorable climatic conditions increase sweating, leading to moist skin and allowing yeast to thrive. This combination raises the likelihood of infections, resulting in discomfort and irritation. *Candida* spp. are commonly found in the soil, hospital environments, on inanimate objects, in food, and as a typical inhabitant of the human body. They can be isolated from diseased skin, the gastrointestinal tract, female genitalia, and indwelling medical devices. Particularly, poorly air-conditioned hospital environments and fragile skin favor the penetration of *Candida* spp. into the body through vascular and urinary devices, favoring bacteremia caused by *Candida* species [28,29].

Our analysis identified several risk factors for candidemia. These include compromised immune systems resulting from conditions such as diabetes (20.6%) and solid organ tumors (20.6%) [30]. Additionally, other significant risk factors were chronic heart failure (34.3%), chronic obstructive pulmonary disease (22.3%), and chronic renal failure (20.0%). The prevalence of these conditions was probably due to the older age of our sample population [31].

Our study found no significant seasonal distribution of *Candida* spp. in hospital settings, as reported in Table 2.

We found a link between death during hospitalization and *C. parapsilosis.* This finding is confirmed by its presence in the WHO priority pathogens list [1]. *C. parapsilosis* is globally distributed and recognized for causing an increasing proportion of invasive *Candida* spp. infections. It is associated with high mortality across all fragile populations and linked to nosocomial outbreaks, especially involving the use of invasive medical devices such as central venous catheters [1,32,33].

In our study, the distribution of *Candida* spp. showed a prevalence of *C. parapsilosis* in ICUs (54.5%), *C. albicans* in medical settings (72.5%), and *C. glabrata* in surgical settings (24.0%).

Emerging studies have confirmed these findings, highlighting the prevalence of *C. parapsilosis* infections among ICU patients, especially for fluconazole-resistant isolates [32,33,34,35].

*C. glabrata* was more frequently isolated after abdominal surgery, and the likelihood of a positive culture for *C. glabrata* increased with patient age. This follows our findings that the patients with *C. glabrata* isolated in their blood had a median age of 73 [36]. Notably, *C. glabrata is* Turkey’s third leading cause of candidemia, as reported by Arastehfar, A. et al. [37]. 

As reported above, we found a significant presence of *C. albicans* in medical settings. This result could be due to the choice to include in the medical settings category several medical units, such as Geriatric, Infectious Disease, Hematology, and Respiratory Units [38,39,40,41].

The analysis of every season showed a similar prevalence of *Candida* species in patient blood samples. Particularly, *C. albicans* and *C. parapsilosis* were the most common isolates for each season, except for Spring, where the most frequent isolates were *C. parapsilosis* and *C. glabrata*.

Recent investigations have identified a significant prevalence of vulvovaginitis caused by non-albicans *Candida* spp., particularly *C. parapsilosis* and *C. glabrata*. In Europe, vulvovaginitis is more common in Spring and Summer. Our findings show a similar trend, with higher isolation rates for these species in Spring [25,42]. In Italy, increased temperatures and humidity during Spring may contribute to skin and mucous membrane infections, especially in the elderly, facilitating the spread of these pathogens via urinary catheters [40].

Moreover, the comparison among seasons of our blood-positive cultures for *Candida* spp. showed seasonal variability among *Candida* spp., especially for *C. glabrata*, with increased rates in Autumn when the temperature decreased. To our knowledge, this is the first report on this association in Mediterranean countries. In the Asia region, Hwang, Seo Young et al. [43] examined the distribution trends of *Candida* spp. in a Korean hospital, their antifungal susceptibilities, and seasonal variations. Potential pathophysiological mechanisms that may explain the variation in Candida distribution in relationship to season include geographical area, changes in temperature, dietary habits, and bodily adaptations due to heightened exposure to sunlight [43,44,45].

Our study found no statistically significant differences in antifungal resistance among the seasons regarding the distribution of *Candida* spp., while in every season, among the tested antifungal drugs, a significant resistance to fluconazole was shown (10.9% in Summer, 6.3% in Winter, 9.7% in Spring, and 12.6% in Autumn, *p* < 0.05) (see Table 3). 

Considering the most frequent *Candida* spp., we found a higher frequency of resistance to fluconazole and amphotericin in the Summer for the *C. parapsilosis* species, with rates of 20% and 23%, respectively (Figure 1). This follows European and Italian epidemiological trends that showed an increase in fluconazole resistance in *Candida* spp. [34,35,36,37,38].

Antifungal resistance in *Candida* species is a serious issue that needs immediate attention. The spread and severity of these infections are affected by the seasons. Rising temperatures from global warming increase resistance and help Candida form biofilms, making infections harder to treat. Certain types of Candida, especially C. tropicalis, are linked to specific seasons and regions where antifungal resistance is extreme, as reported Lima, R. et al. [46].

Antifungal stewardship strategies have been implemented to optimize medication use, including the selection of drugs, dosages, and treatment durations according to international guidelines [47]. Finally, there were no changes in prescription patterns during the study period for fluconazole that could have influenced resistance patterns.

## 4. Limitations

This is a retrospective pilot study conducted at a single center. The study provides preliminary results that need to be confirmed by a multicenter study across different geographic areas and a large sample size, since our statistical analyses may be impacted due to the small sample, and some nonsignificant results were obtained, and some statistical tests not performed.

This preliminary study was designed to reduce costs and was conducted to identify possible correlation between candidemia and seasonality, before conducting a study on a very large sample of patients. Nevertheless, this study focused on some remarkable data associated with seasonality. Since statistical biases could influence the obtained result due to the small sample size, the authors performed a power analysis using effect size. We observed that all the significant tests showed a low probability of statistical biases due to their small sample size.

Finally, regarding any bias introduced during the selection phase of the collection and microbiological analysis of the blood samples, as described in the Section 5, the procedures were carefully conducted.

## 5. Materials and Methods

We performed a retrospective observational cohort study on the inpatient population admitted to wards or the Intensive Care Unit for at least 48 h to the public academic Policlinico University Hospital “Paolo Giaccone (A.U.O.P.)”, Palermo, Italy, a tertiary-level hospital with 600 inpatient beds and nearly 14,000 hospital admissions per year, between 1 January 2022 and 31 December 2024 [48]. We included only adult patients (18 years or older) with *Candida* species in the blood according to previous recommendations [49].

Blood culture samples were collected aseptically via peripheral venipuncture, intravenous catheters (Peripherally Inserted Central Catheter, midline), or central venous catheters (CVCs), and the positive BC results were evaluated for contamination (i.e., false positives). Particularly, the patients with intravenous catheters or central venous catheters positive for *Candida* spp. (28.6% = 50/175) had at least one positive venous peripheral positive for the same *Candida* spp.

All blood samples excluded had a high risk of contamination by nursing practices, such as the reuse of retrograde syringes, total parenteral nutrition, or retrograde medication administration, as reported by Sherertz RJ et al. [50].

Patients’ records included age, sex, isolated candida species, hospital wards, and antifungal susceptibility patterns. The data records were obtained from a database using the microbiological institutional electronic information system of the Hospital Unit for the prevention and control of hospital infection at (A.U.O.P.).

The risk factors for candidemia vary by healthcare setting. The authors classify wards into three categories—medical, surgical, and ICU—reflecting differences in procedures and patient populations. Particularly, the Medical Area included Cardiology, Endocrinology, Gastroenterology, Geriatrics, Hematology, Infectious Diseases, Internal Medicine, Nephrology, Neurology, Oncology, and Pulmonology wards. The Surgical Area included Cardiovascular Surgery, General Surgery, Neurosurgery, Oncological Surgery, Otolaryngology, Orthopedic Surgery, Plastic Surgery, Urology, Vascular Surgery, and Obstetrics and Gynecology. Finally, the Intensive Care Unit included General Intensive Care, the Post-Operative Intensive Care Unit, and Specialized Intensive Care.

*Candida* spp. isolates from blood cultures were analyzed across Winter, Spring, Summer, and Autumn to observe variations in species distribution according to the studied variables. For the seasonal analysis, only the first day of positive candidemia was considered in patients with recurrent or persistent blood candidemia.

The procedures performed in this study followed the ethical standards of the Institutional and/or National Research Committee and the Helsinki Declaration. The study protocol was approved by the Ethics Committee of the Azienda Ospedaliero–Universitaria Policlinico “P. Giaccone”, University of Palermo, Palermo, Italy (IRB n. 28-12/2024).

### 5.1. Microbiological Methods 

The laboratory-based microbiological investigation was performed in the Microbiology and Virology Unit of University Hospital Policlinico “P. Giaccone” in Palermo (Italy) [51,52]. *Candida* spp. were isolated and identified with a MALDI-TOF-MS system (MALDI Biotyper CA System, Bruker Daltonics Inc., Billerica, MA, USA) according to standard procedures [53,54]. The susceptibility testing results considered the materials and methods used for the MIC data and used an interpretive approach with the EUCAST breakpoints and guidelines for MALDI-TOF MS [55,56].

### 5.2. Statistical Analysis 

Categorical variables were described with numbers or percentages, while continuous variables were expressed as mean ± standard deviation (SD), and/or median and interquartile interval (IRQ = [Q1; Q3]).

Multiple comparison chi-square tests were used to individualize significant differences among three or more independent variables. If the chi-square test was significant (*p* < 0.05), a post hoc Z-test was performed to identify the highest or lowest significant frequency. Fisher’s exact test was used if the chi-square test was not appropriate. The chi-square goodness of fit was used to evaluate significant differences among three or more modalities of a variable. The Shapiro–Wilk test was used to check whether the data were normally distributed. The Kruskal–Wallis test, followed by a post hoc test with the Conover test for pairwise comparison, was performed for multi-comparison among three or more independent variables in the case of no normal distribution condition. Cochran’s Q-test was used to evaluate the differences between three or more dependent variables. When the Cochran’s Q-test was positive (*p* < 0.05), a post hoc Bonferroni–Dunn test was applied to individualize significant differences between all possible variables. Additionally, since the sample size was smaller in some analyses, we used effect size to perform power analyses for each significant statistical test. Particularly, the effect sizes were computed by the phi coefficient for categorical variables, by *h*^2^ and *r* for non-parametric tests (Mann–Whitney test and Wilcoxon signed-rank test, respectively), and by Cohen’s *d* index (paired and unpaired *t*-tests).

MATLAB statistical toolbox version 2008 (MathWorks, Natick, MA, USA) was used for data analysis.

#### Seasonal Variable

Our study aimed to evaluate the impact of climate on bloodstream infections caused by *Candida* spp. in Sicily. We considered seasonality as a key variable in our analysis. Environmental parameters such as temperature, atmospheric pressure, precipitation index, etc., were not considered individually, as they are parameters that do not have standard variations but are highly variable. Therefore, the use of these parameters could bring a bias to our statistical analyses. The authors overcame this bias by considering a standard measure of climate variations in Italy, represented by the seasons. This can be considered a macro parameter that involves all the others. Mathematically, this step is the basis of reducing our statistical biases, i.e., identifying a standard and dominant parameter that can include all the other sub-parameters with greater variability.

## 6. Conclusions

The rise in *Candida* spp. cases during Summer can be attributed to increased temperatures and heightened sunlight exposure. To help mitigate the risk of *Candida* spp. bloodstream infections, patients in hospitals who are vulnerable should be proactive about their habits during this season. Understanding the seasonality of environmental contamination is vital for addressing its impact on hospitalized populations. Therefore, exploring the seasonal patterns of these factors and their effects on patients is essential for improving care. Additionally, the growing resistance to fluconazole and the mortality associated with *C. parapsilosis* underscore the need for a robust surveillance system in the ICU. By prioritizing further research, we can develop more effective strategies to tackle this common condition and enhance patient outcomes.

A recent study by Laura García-Gutierrez et al. [57] focused on the environmental conditions within hospitals, with a particular emphasis on air quality. The research aimed to analyze the gradients present in these environments and highlighted the prevalence of certain opportunistic pathogens. It is essential to reconsider key factors that could compromise controlled conditions in sensitive areas, such as HEPA-protected rooms. These factors may include the introduction of organic matter by healthcare personnel via contaminated shoes and clothing, irregular functioning of ventilation systems, and insufficient monitoring of personnel access.

Finally, we suggest the implementation and development of new environmental monitoring programs in hospitals that include air conditioning capable of adapting the internal temperature of the structure to sudden changes in external environmental temperatures to guarantee a reduction in at-risk biological surfaces (skin or mucous membranes) and consequently a decrease in *Candida* spp. bloodstream infections.

## Figures and Tables

**Figure 1 antibiotics-14-00452-f001:**
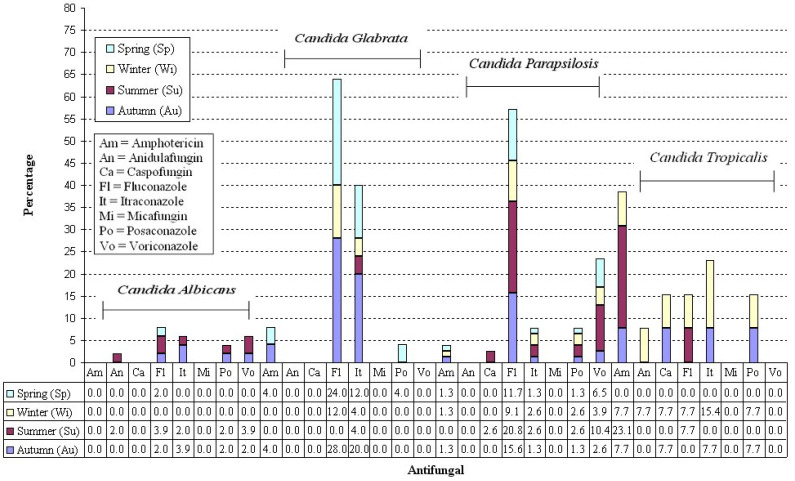
Antibiotic resistance considering season and *Candida* spp. Not including less frequent *Candida* spp. such as *C. krusei*, *C. lusitaniae*, *C. norvegensis*, and *C. pulcherrima*.

**Table 1 antibiotics-14-00452-t001:** General characteristics of 175 patients.

Parameters	Sample
*Patients*	175
*Age at hospitalization*	
Mean ± SD	68.3 (14.2)
Median (IQR)	70.0 (61.3, 77.8)
*Gender*	
Male	57.1% (100)
Female	42.9% (75)
*Deaths*	41.7% (73)
*Hospital setting*	
ICU	37.7% (66)
Medical setting	51.4% (90)
Surgical setting	10.9% (19)
*Comorbidities* †	
Acute coronary syndrome (ACS)	6.9% (12)
Autoimmune diseases (AD)	2.3% (4)
Chronic heart failure (CHF)	34.3% (60)
Chronic obstructive pulmonary disease (COPD)	22.3% (39)
Chronic renal failure (CRF)	20.0% (35)
Dementia (DM)	9.1% (16)
Diabetes	20.6% (36)
Organ solid tumors	20.6% (36)
Hematological tumors	4.6% (8)
Cerebrovascular (brain) insult	13.1% (23)
Haemodialysis (HD)	10.9% (19)
Liver cirrhosis (LC)	4.0% (7)
*Candida* spp. *by Season*	
Autumn	28.6% (50)
Summer	32.0% (56)
Winter	18.3% (32)
Spring	21.1% (37)

SD = standard deviation; IQR= interquartile range; † = some patients had two or more comorbidities.

**Table 2 antibiotics-14-00452-t002:** Percentages of *Candida* spp. considering both hospital setting and season.

Hospital Setting/Seasons	Autumn	Summer	Winter	Spring
ICU	10.9% (19)	11.4% (20)	8.0% (14)	7.4% (13)
Medical	15.4% (27)	17.7% (31)	9.7% (17)	8.6% (15)
Surgical	2.3% (4)	2.9% (5)	0.6% (1)	5.1% (9)

**Table 3 antibiotics-14-00452-t003:** Percentages of *Candida* spp. resistance considering both the type of antifungal and the season.

Antifungal	Total	Autumn (Au)	Summer (Su)	Winter (Wi)	Spring (Sp)	Analysis Among Seasons *p*-Value (Test)
**Total**	175	28.6% (50)	32.0% (56)	18.3% (32)	21.1% (37)	0.0364 * (Cg) Su **, *p* = 0.0263(Z) Effect size: phi = 0.644, large effect
(1) Amphotericin	7.4% (13)	2.9% (5)	2.3% (4)	1.1% (2)	1.1% (2)	0.56 (Cg)
(2) Anidulafungin	1.1% (2)	0.0% (0)	0.6% (1)	0.6% (1)	0.0% (0)	N/A
(3) Caspofungin	2.3% (4)	0.6% (1)	1.1% (2)	0.6% (1)	0.0% (0)	N/A
(4) Fluconazole	39.4% (69)	12.6% (22)	10.9% (19)	6.3% (11)	9.7% (17)	0.29 (Cg)
(5) Itraconazole	13.7% (24)	5.7% (10)	2.3% (4)	2.9% (5)	2.9% (5)	0.30 (Cg)
(6) Micafungin	0.0% (0)	0.0% (0)	0.0% (0)	0.0% (0)	0.0% (0)	N/A
(7) Posaconazole	6.9% (12)	2.3% (4)	1.7% (3)	1.7% (3)	1.1% (2)	0.88 (Cg)
(8) Voriconazole	12.0% (21)	1.7% (3)	5.7% (10)	1.7% (3)	2.9% (5)	0.10 (Cg)
**Analysis in** **season,** ***p*-value (test)**	*p* < 0.001 * (Q) (4)>(1),(2),(3),(5),(6),(7),(8), *p* < 0.05 * (B) Effect size: η^2^ = 0.17, large effect	*p* < 0.001 * (Q) (4)>(1),(2),(3),(5),(6),(7),(8), *p* < 0.05 * (B) Effect size: η^2^ = 0.20, large effect	*p* < 0.001 * (Q) (4)>(1),(2),(3),(5),(6),(7), *p* < 0.05 * (B) Effect size: η^2^ = 0.15, large effect	*p* < 0.001 * (Q) (4)>(1),(2),(3),(6),(7),(8), *p* < 0.05 * (B) Effect size: η^2^ = 0.15, large effect	*p* < 0.001 * (Q) (4)>(1),(2),(3),(5),(6),(7),(8), *p* < 0.05 * (B) Effect size: η^2^ = 0.22, large effect	

* = significant test; ** = modality more frequent; Cg = chi-square goodness of fit; Q = Cochran’s Q-test; B = post hoc Q-test by Bonferroni–Dunn method.

**Table 4 antibiotics-14-00452-t004:** Distribution of *Candida* spp. according to age, gender, patient status, and hospital setting. We excluded less frequent *Candida* spp., such as *Candida norvegensis* and *Candida pulcherrima*, from statistical analysis.

Isolated Strains	Age % (n) Mean (SD) Median (IQR)	Gender Males/Females % (n)	Status (††) Dead (Yes) % (n)	Hospital Setting ICU; Medical; Surgical
*Candida albicans (C.A.)* *(n = 51)*	n = 51 69.6 (13.9) 71 (62, 78.5)	56.9% (29) 43.1% (22)	30.5% (18)	17.6% (9); 72.5% (37); 9.8% (5)
*Candida glabrata (C.G.)* *(n = 25)*	n = 25 70.3 (11.5) 73 (62.8, 77.5)	60% (15) 40% (10)	28.0% (7)	28.0% (7); 48.0% (12); 24.0% (7)
*Candida krusei (C.K.)* *(n = 4)*	n = 4 69.8 (16.0) 66.5 (60, 79.5)	25% (1) 75% (3)	0.0% (0)	25.0% (1); 75.0% (3); 0.0% (0)
*Candida lusitaniae (C.L.)* *(n = 3)*	n = 3 73.7 (10.1) 79 (66.3, 79.8)	66.7% (2) 33.3% (1)	66.7% (2)	66.7% (2); 0.0% (0); 33.3% (1)
*Candida norvegensis (C.N.)* † *(n = 1)*	n = 1 65	100% (1) 0.0% (0)	100% (1)	0.0% (0); 100% (1); 0.0% (0)
*Candida parapsilosis (C.P.)* *(n = 77)*	n = 77 66.1 (15.7) 68 (58.5, 77)	57.1% (44) 42.9% (33)	53.2% (41)	54.5% (42); 39.0% (30); 6.5% (5)
*Candida pulcherrima (C.Pu.)* † *(n = 1)*	n = 1 71	100% (1) 0.0% (0)	0.0% (0)	0.0% (0); 0.0% (0); 100% (1)
*Candida tropicalis (C.T.)* *(n = 13)*	n = 13 70.8 ± 12.9 71 (67.5, 79)	53.8% (7) 46.2% (6)	30.8% (4)	38.5% (5); 53.8% (7); 7.7% (1)
** *p* ** **-value (test)**	*p* = 0.86 (KW)	*p* = 0.88 (F)	*p* < 0.0402 * (F) *C.P*. **, 53.2%, *p* = 0.005 (Z) Effect size: phi = 0.86, large effect	*p* = 0.0001 * (F) C.P./ICU **, 54.5%, *p* = 0.0001(Z) C.A./Medical **, 72.5%, *p* = 0.0003(Z) C.G./Surgical **, 24%, *p* = 0.0161 (Z) Effect size: phi = 2.37, large effect

† = *Candida* species not included in statistical analysis; †† = died during hospitalization * = significant test; F = Fisher’s exact test; Z = post hoc Z-test; ** = more frequent; KW = Kruskal–Wallis test.

**Table 5 antibiotics-14-00452-t005:** *Candida* spp. isolated from 175 patients and stratified for season.

Isolated Strains	Total	Autumn (Au)	Summer (Su)	Winter (Wi)	Spring (Sp)	*p*-Value Among Seasons (Test)
**Total**	175	28.6% (50)	32.0% (56)	18.3% (32)	21.1% (37)	0.053 (Cg)
*Candida albicans* *(C.A.)*	29.1% (51)	8.0% (14)	11.4% (20)	5.1% (9)	4.6% (8)	0.07 (Cg)
*Candida glabrata* *(C.G.)*	14.3% (25)	5.7% (10)	0.6% (1)	2.9% (5)	5.1% (9)	**0.0436 * (Cg)** **Au **, *p* = 0.0481 (Z)** Effect size: phi = 1.62, large effect
*Candida krusei* *(C.K.)*	2.3% (4)	1.7% (3)	0.6% (1)	0.0% (0)	0.0% (0)	N/A
*Candida lusitaniae* *(C.L.)*	1.7% (3)	0.6% (1)	0.0% (0)	0.0% (0)	1.1% (2)	N/A
*Candida norvegensis* *(C.N.)*	0.6% (1)	0.0% (0)	0.6% (1)	0.0% (0)	0.0% (0)	N/A
*Candida parapsilosis* *(C.P.)*	44.0% (77)	10.3% (18)	15.4% (27)	8.6% (15)	9.7% (17)	0.22 (Cg)
*Candida pulcherrima (C.Pu.)*	0.6% (1)	0.0% (0)	0.6% (1)	0.0% (0)	0.0% (0)	N/A
*Candida tropicalis* *(C.T.)*	7.4% (13)	2.3% (4)	2.9% (5)	1.7% (3)	0.6% (1)	0.44 (Cg)
**Analysis into groups** *p*-value (test)	*p* < 0.0001* (Cg) *C.A.* **, *p* < 0.0001(Z) *C.P.* **, *p* < 0.0001(Z) Effect size: phi = 11.5, large effect	*p* < 0.0001 * (Cg) *C. A.* **, *p* = 0.0002 (Z) *C.P.* **, *p* < 0.0001 (Z) Effect size: phi = 2.37, large effect	*p* < 0.0001 * (Cg) *C. A.* **, *p* < 0.0001 (Z) *C.P.* **, *p* < 0.0001 (Z) Effect size: phi = 3.89, large effect	*p* < 0.0001 * (Cg) *C. A.* **, *p* = 0.0027 (Z) *C.P.* **, *p* < 0.0001 (Z) Effect size: phi = 5.61, large effect	*p* < 0.0001 * (Cg) *C.G.* **, *p* = 0.0237 (Z) *C.P.* **, *p* < 0.0001 (Z) Effect size: phi = 5.62, large effect	

N/A = test not performed, n < 10; * = significant test; Cg = chi-square goodness fit; Z = post hoc Z-test; ** = more frequent.

**Table 6 antibiotics-14-00452-t006:** Hospital setting, comorbidities, and *Candida* spp. related to patients with positive blood samples for *Candida* spp. with central venous catheters.

Parameters	Sample
*Central venous catheters*	28.6% (50)
*Hospital setting*	
ICU	38.0% (19)
Medical	44.0% (22)
Surgical	18.0% (9)
*Comorbidities* †	
Chronic heart failure	24.0% (12)
Chronic obstructive pulmonary disease	18.0% (9)
Chronic renal failure	28.0% (14)
Diabetes	16.0% (8)
Oncologic patients †††	30.0% (15)
*Candida* spp. ††	
*Candida albicans*	30.0% (15)
*Candida glabrata*	10.0% (5)
*Candida parapsilosis*	50.0% (25) *
*Candida tropicalis*	10.0% (5)

† = comorbidities more frequent; †† = *Candida* spp. more frequent; ††† = including organ solid tumors and/or hematologic tumors; * = significance percentage obtained using chi-square goodness of fit.

## Data Availability

The dataset is available on request from the project manager, Paola Di Carlo.

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
