# Peer review of "The Influence of the Seasonal Variability of Candida spp. Bloodstream Infections and Antifungal Treatment: A Mediterranean Pilot Study"

_antibiotics, 2025, doi:10.3390/antibiotics14050452_

Round 1
Reviewer 1 Report
Comments and Suggestions for Authors
This manuscript presents a valuable preliminary study on the seasonal variability of Candida spp. bloodstream infections in a university hospital in Sicily. The connection between climate factors and fungal infections is timely and relevant, particularly with increasing concerns about climate change impacts on infectious diseases. The retrospective analysis of 175 patients provides interesting insights into species distribution, antifungal resistance patterns, and relationships with hospital settings.
Specific Comments
Introduction
- Lines 53-58 (p.2): The introduction effectively establishes Candida as a WHO priority pathogen but would benefit from more specific information about the prevalence rates of candidemia in Mediterranean regions compared to other parts of Europe. Consider adding epidemiological data specific to Sicily or Southern Italy.
- Lines 74-79 (p.2): The connection between climate change and fungal infections is mentioned but not fully developed. Strengthen this section with more recent references on how climate change specifically affects Candida species distribution and virulence.
Methods
- Lines 238-243 (p.10): The inclusion criteria for patients are mentioned, but exclusion criteria are not clearly defined. Please specify how potential contaminants or false positives were distinguished from true bloodstream infections.
- Lines 255-261 (p.11): The classification of ward specialties into three areas (Medical, Surgical, and ICU) is provided, but the rationale for this grouping should be explained. For instance, why were Obstetrics and Gynecology included in the Medical Area rather than as a separate category?
- Lines 271-276 (p.11): The microbiological methods section would be strengthened by including details about quality control procedures for species identification and antifungal susceptibility testing. Were any reference strains used to validate the methods?
Results
- Lines 99-102 (p.4) and Table 1: The demographic characteristics are presented, but there is no information about underlying conditions or comorbidities of patients, which are important risk factors for candidemia. Consider adding this information or explaining why it was not included.
- Table 3 (p.5): The presentation of antifungal resistance is difficult to interpret as presented. Consider redesigning this table to more clearly show resistance patterns across seasons and species.
- Figure 1 (p.8): This figure contains valuable information but is extremely difficult to read due to small font size and complex structure. Redesign this figure, possibly splitting it into multiple panels for clarity.
- Lines 145-148 (p.8): The statement about C. glabrata and C. parapsilosis being more frequent in Spring is interesting but the clinical or ecological significance of this finding is not adequately explored in the Discussion.
Discussion
- Lines 210-218 (p.9): The discussion about seasonal variability of Candida spp relies heavily on speculation about temperature and humidity. Consider analyzing or at least discussing actual temperature and humidity data from your region during the study period to strengthen these associations.
- Lines 219-227 (p.10): The statement about fluconazole resistance is important but lacks context about antifungal stewardship practices in your institution. Were there any changes in prescribing patterns during the study period that might have influenced resistance patterns?
- Lines 228-235 (p.10): The limitations section acknowledges the pilot nature of the study but should address potential biases more explicitly, such as selection bias and information bias inherent in retrospective studies.
- Lines 298-307 (p.12): The conclusion section makes reasonable recommendations, but the suggestion about air conditioning systems (lines 308-312) seems to come without supporting evidence from your study. Either provide data that supports this recommendation or frame it as a hypothesis for future research.
The manuscript requires significant language revision throughout. Numerous grammatical errors, awkward phrasing, and syntax issues affect readability and clarity. Inconsistent tense usage and redundant expressions appear frequently. Consider professional English editing services to improve sentence structure and scientific terminology. Particular attention should be paid to the Results and Discussion sections, where complex ideas need clearer expression. Addressing these language issues will substantially enhance the overall quality and impact of this important research.
Author Response
REVIEWER 1
This manuscript presents a valuable preliminary study on the seasonal variability of Candida spp. bloodstream infections in a university hospital in Sicily. The connection between climate factors and fungal infections is timely and relevant, particularly with increasing concerns about climate change impacts on infectious diseases. The retrospective analysis of 175 patients provides interesting insights into species distribution, antifungal resistance patterns, and relationships with hospital settings.
Specific Comments
Introduction
1) Lines 53-58 (p.2): The introduction effectively establishes Candida as a WHO priority pathogen but would benefit from more specific information about the prevalence rates of candidemia in Mediterranean regions compared to other parts of Europe. Consider adding epidemiological data specific to Sicily or Southern Italy.
[REPLY]: Thank you for your question. We added the lines 59-65, where we describe status of candidemia in Europe about the most frequent Candida species.
2) Lines 74-79 (p.2): The connection between climate change and fungal infections is mentioned but not fully developed. Strengthen this section with more recent references on how climate change specifically affects Candida species distribution and virulence.
[REPLY]: Thank you for your suggestion. We added the lines 87-93 and references
- Pierantoni, D. C., Corte, L., Casadevall, A., Robert, V., Cardinali, G., & Tascini, C. How does temperature trigger biofilm adhesion and growth in Candida albicans and two non‐Candida albicans Candida species? Mycoses, 64(11), 1412. https://doi.org/10.1111/myc.13291
- Nawaz, Ayman and Khadija, Bibi and Saqib, Muhammad Arif Nadeem and Tara, Tanveer and Umar, Maaz, Unveiling the Impact of Atmospheric Temperature on Antifungal Resistance and Virulence Factors in Candida Spp. Isolated from Forest Ecosystem. Available at SSRN: https://ssrn.com/abstract=5218883 or http://dx.doi.org/10.2139/ssrn.5218883 (2025).
- No authors’ list. High time to tackle drug-resistant fungal infections. Nature. 2025 Apr;640(8059):569. doi: 10.1038/d41586-025-01177-x. PMID: 40234582.
Methods
1) Lines 238-243 (p.10): The inclusion criteria for patients are mentioned, but exclusion criteria are not clearly defined. Please specify how potential contaminants or false positives were distinguished from true bloodstream infections.
[REPLY]: Thank for your suggestion. The following sentences was changed as
Blood culture samples were collected aseptically via peripheral venipuncture or intravenous catheter (Peripherally Inserted Central Catheter, midline), or from central venous catheters (CVC), and the positive BC results were evaluated for contamination (i.e., false positive). Particularly, the patients with intravenous catheter or central venous catheters positive to Candida spp, had at least one positive venous peripheral positive of same Candida spp.
All blood samples excluded had a high risk of contamination by nursing practices, such as reuse of the retrograde syringes, total parenteral nutrition, and retrograde medication administration as reported by Sherertz RJ et al.,, [1]
- Sherertz RJ, Gledhill KS, Hampton KD, Pfaller MA, Givner LB, Abramson JS, Dillard RG. Outbreak of Candida bloodstream infections associated with retrograde medication administration in a neonatal intensive care unit. J Pediatr. 1992 Mar;120(3):455-61. doi: 10.1016/s0022-3476(05)80920-5. PMID: 1538298.
2) Lines 255-261 (p.11): The classification of ward specialties into three areas (Medical, Surgical, and ICU) is provided, but the rationale for this grouping should be explained. For instance, why were Obstetrics and Gynecology included in the Medical Area rather than as a separate category?
[REPLY]: Thank you for your suggestion. The Obstetrics and Gynaecology Unit is now included in the Surgical setting.
3) Lines 271-276 (p.11): The microbiological methods section would be strengthened by including details about quality control procedures for species identification and antifungal susceptibility testing. Were any reference strains used to validate the methods?
[REPLY]: Thank you for your question. Laboratory microbiological investigation was performed at Microbiology and Virology Unit of University Hospital Policlinico "P. Giaccone" in Palermo (Italy). Candida spp were isolated and identified with MALDI-TOF-MS system (MALDI Biotyper CA System, Bruker Daltonics Inc., USA) according to standard procedures [1,2]. The susceptibility testing results consider the materials and methods used for the MIC data and the interpretive approach in accordance with EUCAST breakpoints and guidelines for MALDI-TOF MS [3,4]
- Lacroix C, Gicquel A, Sendid B, Meyer J, Accoceberry I, François N, Morio F, Desoubeaux G, Chandenier J, Kauffmann-Lacroix C, Hennequin C, Guitard J, Nassif X, Bougnoux ME. Evaluation of two matrix-assisted laser desorption ionization-time of flight mass spectrometry (MALDI-TOF MS) systems for the identification of Candida species. Clin Microbiol Infect. 2014 Feb;20(2):153-8. doi: 10.1111/1469-0691.12210. Epub 2013 Apr 17. PMID: 23594150.
- Wilson DA, Young S, Timm K, Novak-Weekley S, Marlowe EM, Madisen N, Lillie JL, Ledeboer NA, Smith R, Hyke J, Griego-Fullbright C, Jim P, Granato PA, Faron ML, Cumpio J, Buchan BW, Procop GW. Multicenter Evaluation of the Bruker MALDI Biotyper CA System for the Identification of Clinically Important Bacteria and Yeasts. Am J Clin Pathol. 2017 Jun 1;147(6):623-631. doi: 10.1093/ajcp/aqw225. PMID: 28505220.
- Astvad KMT, Arikan-Akdagli S, Arendrup MC. A Pragmatic Approach to Susceptibility Classification of Yeasts without EUCAST Clinical Breakpoints. J Fungi (Basel). 2022 Jan 30;8(2):141. doi: 10.3390/jof8020141. PMID: 35205895; PMCID: PMC8877802.
- European Committee on Antimicrobial Susceptibility Testing Breakpoint tables for interpretation of MICs for antifungal agents Available onlinehttps://www.eucast.org/fileadmin/src/media/PDFs/EUCAST_files/AFST/Clinical_breakpoints/AFST_BP_v11.0.pdf (accessed on 03 April 2024).
Results
1) Lines 99-102 (p.4) and Table 1: The demographic characteristics are presented, but there is no information about underlying conditions or comorbidities of patients, which are important risk factors for candidemia. Consider adding this information or explaining why it was not included.
[REPLY]: Thank you for your question. We added the comorbidities in Table 1 and we added the following sentences in the discussion section and references
Our analysis identified several risk factors for candidemia. These include com-promised immune systems resulting from conditions such as diabetes (20.6%) and solid organ tumors (19.4 %) . Additionally, other significant risk factors were chronic heart failure (34.3%), chronic obstructive pulmonary disease (22.3%), and chronic renal failure (20.0%). The prevalence of these conditions was notable due to the older age of our sample population.
- Wolfgruber, S., Sedik, S., Klingspor, L., Tortorano, A., R Gow, N. A., Lagrou, K., Gangneux, P., Maertens, J., Meis, J. F., Lass-Flörl, C., Arikan-Akdagli, S., Cornely, O. A., & Hoenigl, M. Insights from Three Pan-European Multicentre Studies on Invasive Candida Infections and Outlook to ECMM Candida IV. Mycopathologia, 189(4), 70. https://doi.org/10.1007/s11046-024-00871-0
- Barchiesi F, Orsetti E, Mazzanti S, Trave F, Salvi A, Nitti C, Manso E. Candidemia in the elderly: What does it change? PLoS One. 2017 May 11;12(5):e0176576. doi: 10.1371/journal.pone.0176576. PMID: 28493896; PMCID: PMC5426612.
2) Table 3 (p.5): The presentation of antifungal resistance is difficult to interpret as presented. Consider redesigning this table to more clearly show resistance patterns across seasons and species.
[REPLY]: Thank you for your question. We consider Table 3 more useful in this form, therefore a more detailed explanation was provided for it. Additionally, we recalculated all percentages on the total sample (n=175), to facilitate interpretation, since two analyses were performed on the same table.
3) Figure 1 (p.8): This figure contains valuable information but is extremely difficult to read due to small font size and complex structure. Redesign this figure, possibly splitting it into multiple panels for clarity.
[REPLY]: Thank you for your suggestion. A more detailed explanation was provided for Figure 1. Additionally, we have increased the font size.
4) Lines 145-148 (p.8): The statement about C. glabrata and C. parapsilosis being more frequent in Spring is interesting but the clinical or ecological significance of this finding is not adequately explored in the Discussion.
[REPLY]: Thank you for your question. This result was due by statistical analysis into every season. Particularly apart C. parapsilosis that was more frequent in every season (please see the statistical results in last row of Table 5), C. glabrata was more frequent only in Spring. We discuss it at lines 203-209 (original paper) in the Discussion section. The result was in accordance to references 26, 40
Discussion
1) Lines 210-218 (p.9): The discussion about seasonal variability of Candida spp relies heavily on speculation about temperature and humidity. Consider analyzing or at least discussing actual temperature and humidity data from your region during the study period to strengthen these associations.
[REPLY]: Thank you for your question. We clarified the in subsection 5.1.1 of Methods the use of the variable seasonality and of the sub variables such as temperature and humidity.
2) Lines 219-227 (p.10): The statement about fluconazole resistance is important but lacks context about antifungal stewardship practices in your institution. Were there any changes in prescribing patterns during the study period that might have influenced resistance patterns?
[REPLY]: Thank you for your comment. We added the lines 242-246.
According to hospital guidelines, only infectious disease specialists can prescribe antifungals. Antifungal medications are prescribed exclusively in cases of candidemia or when a severe infection by Candida spp. is confirmed by sterile microbiological specimens. The selection of the appropriate drug, dosage, and duration of treatment must follow established hospital guidelines.
If a microbiological sample is not sterile, source control measures—such as replacing urinary, venous, and other devices—are implemented. Antifungals are not prescribed for respiratory tract samples if Candida spp. is isolated.
Additionally, before any hospital admission, we assess the history of previous antifungal treatments, particularly the commonly used fluconazole, to monitor any potential increase in resistance among Candida spp.
Antifungal stewardship strategies were implemented to optimize medication use with the selection of the drug, dosage and treatment duration according to international guidelines.
Finally, for fluconazole, there were no changes in prescription patterns during the study period that could have influenced resistance patterns.
3) Lines 228-235 (p.10): The limitations section acknowledges the pilot nature of the study but should address potential biases more explicitly, such as selection bias and information bias inherent in retrospective studies.
[REPLY]: Thank you for your question. The section is improved.
4) Lines 298-307 (p.12): The conclusion section makes reasonable recommendations, but the suggestion about air conditioning systems (lines 308-312) seems to come without supporting evidence from your study. Either provide data that supports this recommendation or frame it as a hypothesis for future research.
[REPLY]: Thank you for your question. We improved/added the sentences at lines 389-401
“A recent study by Laura García-Gutierrez et al., [1] focused on the environmental conditions within hospitals, with a particular emphasis on air quality. The research aimed to analyze the gradients present in these environments and highlighted the prevalence of certain opportunistic pathogens concerning. It is essential to reconsider key factors that could compromise controlled conditions in sensitive areas, such as HEPA-protected rooms. These factors may include the introduction of organic matter by healthcare personnel via contaminated shoes and clothing, irregular functioning of ventilation systems, and insufficient monitoring of personnel access.
Finally, we suggest the implementation and developing of new environmental monitoring programs in hospitals that include air conditioning capable of adapting the internal temperature of the structure to sudden changes in external environmental temperatures, to guarantee a reduction in biological surfaces (skin or mucous membranes) at risk and consequently a decrease in Candida spp bloodstream infections.”
- García-Gutiérrez, L., Baena Rojas, B., Ruiz, M., Hernández Egido, S., Ruiz-Gaitán, A. C., Laiz, L., Pemán, J., Cuétara-García, M. S., Mellado, E., & Martin-Sanchez, P. M. (2025). Fungal burden assessment in hospital zones with different protection degrees. Building and Environment, 269, 112454. https://doi.org/10.1016/j.buildenv.2024.112454
Comments on the Quality of English Language
The manuscript requires significant language revision throughout. Numerous grammatical errors, awkward phrasing, and syntax issues affect readability and clarity. Inconsistent tense usage and redundant expressions appear frequently. Consider professional English editing services to improve sentence structure and scientific terminology. Particular attention should be paid to the Results and Discussion sections, where complex ideas need clearer expression. Addressing these language issues will substantially enhance the overall quality and impact of this important research.
[REPLY]: Thank you for your suggestion. We performed an accurate check of English language.
Reviewer 2 Report
Comments and Suggestions for Authors
Thanks for submitting this paper. Good concept but little more co variates could have been evaluated which would have affected the fungal infections. Also kindly find my comments in paper.

Many typo errors and english words needs to be adjusted.
Author Response
REVIEWER 2
Thanks for submitting this paper. Good concept but little more co variates could have been evaluated which would have affected the fungal infections. Also kindly find my comments in paper.
1) Abstract (line 37): “were enrolled…” In prospective studies usually enrolled makes sense in retrospective review Were included better word.
[REPLY]: thank you for your suggestion. We changed
2) Abstract (lines 42-44): Can be rephrased with clear meaning the sentence:
“From the seasonal trend analysis, C. parapsilosis and C. albicans were more frequent for 42 each season, a part for Spring, where the most frequent isolates were C glabrata (5.1%, 43 p=0.0237) and C. parapsilosis (9.7%, p<0.0001).”
[REPLY]: thank you for your suggestion. We rephrased
3) Introduction (line 84). Doubled same words. “ environmentsurrounding environment”
[REPLY]: thank you for your suggestion. We corrected
4) Introduction (line 85). same doubling, “Influences the growth of these biofilms.”
[REPLY]: thank you for your suggestion. We corrected
5) Line 128: hospital
[REPLY]: thank you for your suggestion. We corrected
6) Line 147: apart
[REPLY]: thank you for your suggestion. We corrected
7) Discussion (line:187) IN ICU settings the candidemia and IC has more reasons just than the Seasons association in form of invasive device usage and also the broadspectrum antibiotics usage for longer duration predisposes for IC and also TPN usage is well known risk factor. These earlier established risk factors could have been evaluated to rule out their association in addition to seasonal variations.
[REPLY]: Thank you for your question. Most of the patients included in our study had at least one risk factor indicated by the Reviewer. These risk factors were among the causes of candidemia in hospitalized patients. Therefore, the authors added in Table 1 all the comorbidities identified in our study. Furthermore, the authors emphasize that blood samples obtained from central venous catheters were sampled in a standardized manner to avoid possible contamination.
Finally, the authors emphasize to the Reviewer that our study focuses on the impact of seasonality on candidemia and does not have among its objectives that of evaluating the impact of risk factors on candidemia.
8) Discussion (line:195) Increasing age with more comorbidities and particularly Long standing Diabetes mellitus or History of immunosuppression in oncology patients predisposes them for higher fungal infections. Nowhere in analysis this is accounted for...
[REPLY]: Thank you for your question. We added the comorbidities in Table 1 and we added the following sentences in the discussion section and references
Our analysis identified several risk factors for candidemia. These include com-promised immune systems resulting from conditions such as diabetes (20.6%) and solid organ tumors (19.4 %) . Additionally, other significant risk factors were chronic heart failure (34.3%), chronic obstructive pulmonary disease (22.3%), and chronic renal failure (20.0%). The prevalence of these conditions was notable due to the older age of our sample population.
- Wolfgruber, S., Sedik, S., Klingspor, L., Tortorano, A., R Gow, N. A., Lagrou, K., Gangneux, P., Maertens, J., Meis, J. F., Lass-Flörl, C., Arikan-Akdagli, S., Cornely, O. A., & Hoenigl, M. Insights from Three Pan-European Multicentre Studies on Invasive Candida Infections and Outlook to ECMM Candida IV. Mycopathologia, 189(4), 70. https://doi.org/10.1007/s11046-024-00871-0
- Barchiesi F, Orsetti E, Mazzanti S, Trave F, Salvi A, Nitti C, Manso E. Candidemia in the elderly: What does it change? PLoS One. 2017 May 11;12(5):e0176576. doi: 10.1371/journal.pone.0176576. PMID: 28493896; PMCID: PMC5426612.
9) Discussion (line:208), Use of invasive devices like central catheters and urinary catheters and drains and its association to invasive fungal infections is to be ascertained?
[REPLY]: Thank you for your question. We added the Table 6 and in the Discussion section please see the lines 224 -228
“C. parapsilosis is globally distributed and recognized for causing an increasing proportion of invasive Candida spp infections. It is associated to high mortality across all fragile populations and linked to nosocomial outbreaks, especially involving the use of invasive medical devices such as central venous catheters. [1,31,32].”
10) Also, the ICU population proportion of ventilated versus non ventilated patients also is important as Presence of Endo tracheal tube increases chances of isolation of candida species in tracheal samples.
[REPLY]: Thank you for your question. Based on Table 2, our aim was to describe the likely relationship between hospital setting and seasonality, in relation to candidemia. Therefore, despite this is an important question for patients in ICU, it is not the focus of our paper.
11) Materials and Methods (line 236), usually materials and methods explained before the results
[REPLY]: thank you for your question. We have adopted the format/template of the Antibiotics journal
12) Discussion (line:208), at least
[REPLY]: thank you for your suggestion. We corrected
Reviewer 3 Report
Comments and Suggestions for Authors
Authors described the prevalence of Candida spp bloodstream infections (BSIs) and antifungal susceptibility. They explored the seasonality of Candida infection in Italy by retrospective survey of clinical data from Jan. 2022 to Dec.2024.
Line 272-276 “Candida species were isolated and identified by morphology and MALDI-TOF-MS---------according to EUCAST breakpoints” It should be rephrased.
Table 3 has less meaning and delete. The detection of Candida spp. varies depending on the season, so that antifungal susceptibility might be changed depending on identified Candida spp.
Antifungal susceptibility should be described according to Candida spp. NO meaning of seasonality in antifungal susceptibility.
Line 298-299, the sentence of “The rise in Candida spp cases summer can be attributed to increased temperatures and heightened sunlight exposure” what does it mean? Check the accuracy.
Line 308-312, Check the accuracy of the statement and correct appropriately.
In general, 175 Candida were identified by MALTI-TOF-MS, and antifungal susceptibility is not described specifically for individual Candida spp.
Manuscript is lengthy by unnecessary tables and vague description, containing less original data.
The article should be described precisely and reported as a short communication.
Author Response
REVIEWER 3
Authors described the prevalence of Candida spp bloodstream infections (BSIs) and antifungal susceptibility. They explored the seasonality of Candida infection in Italy by retrospective survey of clinical data from Jan. 2022 to Dec.2024.
1) Line 272-276 “Candida species were isolated and identified by morphology and MALDI-TOF-MS---------according to EUCAST breakpoints” It should be rephrased.
[REPLY]: Thank you for your question. The microbiological section was revised as follows and some references were added:
Laboratory microbiological investigation was performed at Microbiology and Virology Unit of University Hospital Policlinico "P. Giaccone" in Palermo (Italy). Candida spp were isolated and identified with MALDI-TOF-MS system (MALDI Biotyper CA System, Bruker Daltonics Inc., USA) according to standard procedures [1,2]. The susceptibility testing results consider the materials and methods used for the MIC data and the interpretive approach in accordance with EUCAST breakpoints and guidelines for MALDI-TOF MS [3,4]
- Lacroix C, Gicquel A, Sendid B, Meyer J, Accoceberry I, François N, Morio F, Desoubeaux G, Chandenier J, Kauffmann-Lacroix C, Hennequin C, Guitard J, Nassif X, Bougnoux ME. Evaluation of two matrix-assisted laser desorption ionization-time of flight mass spectrometry (MALDI-TOF MS) systems for the identification of Candida species. Clin Microbiol Infect. 2014 Feb;20(2):153-8. doi: 10.1111/1469-0691.12210. Epub 2013 Apr 17. PMID: 23594150.
- Wilson DA, Young S, Timm K, Novak-Weekley S, Marlowe EM, Madisen N, Lillie JL, Ledeboer NA, Smith R, Hyke J, Griego-Fullbright C, Jim P, Granato PA, Faron ML, Cumpio J, Buchan BW, Procop GW. Multicenter Evaluation of the Bruker MALDI Biotyper CA System for the Identification of Clinically Important Bacteria and Yeasts. Am J Clin Pathol. 2017 Jun 1;147(6):623-631. doi: 10.1093/ajcp/aqw225. PMID: 28505220.
- Astvad KMT, Arikan-Akdagli S, Arendrup MC. A Pragmatic Approach to Susceptibility Classification of Yeasts without EUCAST Clinical Breakpoints. J Fungi (Basel). 2022 Jan 30;8(2):141. doi: 10.3390/jof8020141. PMID: 35205895; PMCID: PMC8877802.
- European Committee on Antimicrobial Susceptibility Testing Breakpoint tables for interpretation of MICs for antifungal agents Available onlinehttps://www.eucast.org/fileadmin/src/media/PDFs/EUCAST_files/AFST/Clinical_breakpoints/AFST_BP_v11.0.pdf (accessed on 03 April 2024).
2) Table 3 has less meaning and delete. The detection of Candida spp. varies depending on the season, so that antifungal susceptibility might be changed depending on identified Candida spp.
[REPLY]: Thank you for your comment. We believe this table is very informative for future analysis. The table shows the simultaneous impact of seasonality and antifungal used on resistance by Candida species. The table shows some significant results. We found a significant more resistance to Candida spp in Summer (32%) than Autumn (28.6%), Spring (21.1%), and Winter (18.3%). In addition, from the statistical analysis by column, in each season the Fluconazole was the antifungal with the highest percentage of resistance to Candida spp
3) Antifungal susceptibility should be described according to Candida spp. NO meaning of seasonality in antifungal susceptibility.
[REPLY]: Thank you for your suggestion. Figure 1 shows the antifungal resistances for more frequent Candida spp. Additionally, the following sentence was added and related reference (lines 272-277)
Antifungal resistance in Candida species is a growing concern, with seasonality potentially playing a role in the spread and severity of infections. Higher temperatures, particularly due to global warming, may contribute to increased resistance and enhanced biofilm formation in Candida, making infections more difficult to treat. Certain Candida spp, like C. tropicalis, have been linked to specific seasons and geographical locations, and antifungal resistance is particularly prevalent in some of these regions, according to Lima, R et al.,.
- Lima, R., Ribeiro, F. C., Colombo, A. L., & De Almeida, J. N. (2022). The emerging threat antifungal-resistant Candida tropicalis in humans, animals, and environment. Frontiers in Fungal Biology, 3, 957021. https://doi.org/10.3389/ffunb.2022.957021
4) Line 298-299, the sentence of “The rise in Candida spp cases summer can be attributed to increased temperatures and heightened sunlight exposure” what does it mean? Check the accuracy.
[REPLY]: Thank you for your suggestion. We improved the sentence
5) Line 308-312, Check the accuracy of the statement and correct appropriately.
[REPLY]: Thank you for your suggestion. The sentences at lines 389-401 were improved/added
6) In general, 175 Candida were identified by MALTI-TOF-MS, and antifungal susceptibility is not described specifically for individual Candida spp.
[REPLY]: Thank you for your question. The microbiological section was revised as follows and the references were added,
The susceptibility testing results consider the materials and methods used for the MIC data and the interpretive approach in accordance with EUCAST breakpoints and guidelines for MALDI-TOF MS [1,2]
- Astvad KMT, Arikan-Akdagli S, Arendrup MC. A Pragmatic Approach to Susceptibility Classification of Yeasts without EUCAST Clinical Breakpoints. J Fungi (Basel). 2022 Jan 30;8(2):141. doi: 10.3390/jof8020141. PMID: 35205895; PMCID: PMC8877802.
- European Committee on Antimicrobial Susceptibility Testing Breakpoint tables for interpretation of MICs for antifungal agents Available onlinehttps://www.eucast.org/fileadmin/src/media/PDFs/EUCAST_files/AFST/Clinical_breakpoints/AFST_BP_v11.0.pdf (accessed on 03 April 2024).
Additionally, about antifungal resistance for individual Candida spp, also considering the seasonally, were reported in Figure 1
7) Manuscript is lengthy by unnecessary tables and vague description, containing less original data.
[REPLY]: Thank you for your question. In this comment the Reviewer 3 was very severe. Furthermore, this article provides many important results in the field of the control of Candida infections in hospitalized patients. We emphasize that in the Mediterranean area there are no studies on this topic. Particularly, a recent paper by Gacìa-Gutiérrez L., (2025) has performed a study on all fungi (including Candida) present in the hospital environmental, showing that the seasonal influence of outdoor fungal concentration on fungal contamination in the inside hospitals. This study confirmed some our results obtained on blood human sample.
- García-Gutiérrez, L., Baena Rojas, B., Ruiz, M., Hernández Egido, S., Ruiz-Gaitán, A. C., Laiz, L., Pemán, J., Cuétara-García, M. S., Mellado, E., & Martin-Sanchez, P. M. (2025). Fungal burden assessment in hospital zones with different protection degrees. Building and Environment, 269, 112454. https://doi.org/10.1016/j.buildenv.2024.112454
Regarding the tables included in the paper, in our opinion, it seems that the Reviewer 3 gave a very summary opinion, without considering the many analyses and results described in the tables.
Finally, to verifying whether our results were affected by statistical bias, we also performed a power analysis.
8) The article should be described precisely and reported as a short communication.
[REPLY]: Thank you for your question. It seems that the reviewer's suggestion is to change the article in a short communication. In this case, the authors emphasize that this is an original study that addresses both the seasonality of different Candida species and the resistance to different antifungals. Additionally, the article has been improved after changes made with the reviewers' comments. We thank the Reviewer, but the authors prefer to keep this type of article.
Round 2
Reviewer 1 Report
Comments and Suggestions for Authors
I've reviewed the revised manuscript titled "Influence of Season Variability of Candida spp Bloodstream Infections and Antifungal Treatment: A Mediterranean Pilot Study." The authors have made substantial improvements throughout the manuscript, addressing previous concerns adequately.
The study provides valuable insights into the seasonal distribution of Candida species and antifungal susceptibility in Sicily, with particularly interesting findings regarding C. parapsilosis in ICU settings, C. albicans in medical settings, and C. glabrata in surgical settings. The seasonal variations observed, especially the higher presence of C. glabrata in Autumn and both C. glabrata and C. parapsilosis in Spring, contribute meaningful data to understanding fungal epidemiology in the Mediterranean region.
The manuscript now presents a more coherent analysis of antifungal resistance patterns, particularly for fluconazole, which appears to be the antifungal with most resistance across all seasons. The methodological descriptions have been clarified, statistical analyses are sound, and the addition of Table 6 regarding central venous catheters strengthens the clinical relevance of the findings.
The revised discussion section effectively contextualizes these findings within the broader literature on climate change and fungal disease patterns, and the authors have improved the organization and clarity of their data presentation throughout.
The manuscript is now suitable for publication.
Reviewer 3 Report
Comments and Suggestions for Authors
Appropriate additions and corrections were made.
No more comments from this reviewer.